# Liquid Film Translocation Significantly Enhances Nasal Spray Delivery to Olfactory Region: A Numerical Simulation Study

**DOI:** 10.3390/pharmaceutics13060903

**Published:** 2021-06-18

**Authors:** Xiuhua April Si, Muhammad Sami, Jinxiang Xi

**Affiliations:** 1Department of Aerospace, Industrial, and Mechanical Engineering, California Baptist University, Riverside, CA 92504, USA; asi@calbaptist.edu; 2ANSYS Inc., Houston, TX 77094, USA; Muhammad.Sami@ansys.com; 3Department of Biomedical Engineering, University of Massachusetts, Lowell, MA 01854, USA

**Keywords:** nasal spray, olfactory dose, droplet deposition, liquid film translocation, nose-to-brain delivery, vertex-to-floor position

## Abstract

Previous in vivo and ex vivo studies have tested nasal sprays with varying head positions to enhance the olfactory delivery; however, such studies often suffered from a lack of quantitative dosimetry in the target region, which relied on the observer’s subjective perception of color changes in the endoscopy images. The objective of this study is to test the feasibility of gravitationally driven droplet translocation numerically to enhance the nasal spray dosages in the olfactory region and quantify the intranasal dose distribution in the regions of interest. A computational nasal spray testing platform was developed that included a nasal spray releasing model, an airflow-droplet transport model, and an Eulerian wall film formation/translocation model. The effects of both device-related and administration-related variables on the initial olfactory deposition were studied, including droplet size, velocity, plume angle, spray release position, and orientation. The liquid film formation and translocation after nasal spray applications were simulated for both a standard and a newly proposed delivery system. Results show that the initial droplet deposition in the olfactory region is highly sensitive to the spray plume angle. For the given nasal cavity with a vertex-to-floor head position, a plume angle of 10° with a device orientation of 45° to the nostril delivered the optimal dose to the olfactory region. Liquid wall film translocation enhanced the olfactory dosage by ninefold, compared to the initial olfactory dose, for both the baseline and optimized delivery systems. The optimized delivery system delivered 6.2% of applied sprays to the olfactory region and significantly reduced drug losses in the vestibule. Rheological properties of spray formulations can be explored to harness further the benefits of liquid film translocation in targeted intranasal deliveries.

## 1. Introduction

Nose-to-brain (N2B) drug delivery has been an attractive alternative in neurological disorder therapeutics [1,2,3,4]; however, there is limited success in N2B due to challenges in delivering clinically significant doses to the olfactory region with conventional inhalation devices [5,6,7]. The poor bioavailability to the olfactory mucosa comes from the nonproportionally low ventilation rate to this region and the convoluted nasal structure that filters out most of the inhaled aerosol medications well before they reach the olfactory region [8]. The nasal valve is the narrowest part of the nose and will filter out most inhaled spray droplets. For droplets that have passed the nasal valve, most of them will deposit in the labyrinthic turbinate region. The cross-sectional area expands when the airway transits from the nasal valve to the turbinate, which decreases the airflow speed and promotes large droplets to deposit in the inferior and middle meatuses. The convoluted narrow pathway in the turbinate region also favors droplet deposition via direct interception and inertia impaction. The superior meatus is located at the top of the nose and has a slenderer pathway; therefore, only a very low fraction of inhaled air and droplets can reach the olfactory region that is located at the uppermost region of the nose. Previous studies have demonstrated that less than 1% of applied aerosol medications reached the olfactory region [9]. Using a standard radial-hole inhaler with 140 mL per actuation, González-Botas et al. [10] observed dominant deposition in the nasal valve and inferior meatus, but nearly no deposition at the olfactory cleft. This extremely low bioavailability has prevented nose-to-brain delivery in clinical applications and acts as the bottleneck for the development of new neurological medications [11].

Many efforts have been undertaken to increase the olfactory dosing from nasal sprays by exploring operating factors such as release orientation, spray plume angle, aerosol diameter, and exiting speed [12,13,14,15]. By comparing the deposition distributions in a nasal airway cast from different spray pumps, Cheng et al. [12] reported that a narrow spray plume and smaller droplet diameters allowed more droplets to pass the nasal valve. Kundoor and Dalby [13] investigated the influences of the nozzle orientation in the range of 0–90° and observed that the optimal olfactory dispensing occurred at 60° to 75°. Wang et al. [14] tested the feasibility of intubating the spray nozzle to the middle or superior meatus and actuating the nasal spray right beneath the olfactory region. However, tissue injury risks exist during intubation procedures. This and similar invasive strategies did not gain popularity despite their potential to improve olfactory dosing. More user-friendly strategies were tested by Gizurarson [15], who explored high-speed sprays with narrow plume angles to enhance the transport of spray droplets to the olfactory mucosa but reported limited improvements.

Numerical simulations have been extensively used to investigate nasal spray deliveries [16,17,18,19,20,21,22,23,24,25]. Kimbell et al. [26] simulated the deposition fractions of 20 and 50 μm droplets in an MRI-based nasal airway model with an inhalation rate of 15 L/min and predicted more than 90% of droplets being deposited in the anterior nose. Frank et al. [27] compared intranasal spray penetration in airway models before and after nasal surgery and found that surgical correction of nasal anatomic deformities enhanced spray penetration. Basu et al. [16] numerically tested the effect of reorienting the spray axis and predicted an eightfold improvement in topical delivery of drugs at diseased sites. Tong et al. [23] numerically compared the spray nozzle orientation on intranasal dose distribution and recommended a middle spray direction over the upper or lower directions. Shang et al. [22] simulated the nasal spray drug absorption after deposition.

Compared to extensive experimental and numerical studies of the deposition distribution of spray droplets, only a few studies have investigated the subsequent translocation after the spray droplets are deposited on the nasal airway walls. Using methylene blue dye and nasal endoscopy, Merkus et al. [28] compared four different head positions, i.e., upright, lying head back (Mygind), lying on one side (Kaiteki), and with the head vertex to the floor (vertex-to-floor position, or Mecca). They reported that the vertex-to-floor position delivered more doses to the upper nose and recommended this position for nasal spray delivery to the olfactory region in patients with nasal polyposis [28]. Similarly, by applying three drops of dexamethasone to the nasal vestibule in patients with endoscopic sinus surgery, Cannady et al. [29] observed that the vertex-to-floor position consistently delivered nasal drops to the maxillary sinus (MS), ethmoid cavity, sphenoid sinus, and olfactory cleft. In addition, maintaining the vertex-to-floor position for 5 min led to higher olfactory doses than for 1 min [29], indicating that the liquid film translocation could occur within 5 min after applying the nasal drop. Mori et al. [30] tested the Kaiteki position (lying on the side with the head titled and chin lifted) in healthy subjects and reported that nasal spray could reach the olfactory region in 96% of the decongested subjects and 75% of the untreated cases. Milk et al. [31] compared the administration of nasal drops to the olfactory region between the lying-head-back (Mygind) position and the vertex-to-floor (Mecca) position and reported comparable doses from these two head positions. In the above studies, different methods, such as isotopes, dyes, endoscopy, and pledget collection, have been used to visualize the intranasal distribution of nasal spray distributions; this variability makes the comparison between studies difficult [32]. In addition, subregional doses were often evaluated subjectively by the observers; this qualitative, rather than quantitative, nature of the studies can further confound the intergroup comparison [33]. Furthermore, to develop the dose–response correlation in topical therapeutics for either olfactory impairment or neurological disorders, accurate quantitative knowledge of the olfactory dose is essential.

The objective of this study is to numerically test the hypothesis that the gravitationally driven liquid film migration after spray application can significantly improve drug delivery to the olfactory region. Particularly, the olfactory doses from nasal sprays with a vertex-to-floor head position are quantified in an adult nasal airway model. There are five specific aims hereof as follows:(1)Develop a nasal spray-releasing model with controlled insert position, application orientation, droplet size distribution, exiting velocity distribution, and spray plume angle;(2)Develop an Eulerian-wall-film-based deposition and translocation model for nasal spray applications;(3)Optimize the delivery system in terms of the release position, application orientation, droplet sizes, and plume angle;(4)Simulate the dynamic wall film formation and migration with the baseline and optimized delivery systems;(5)Compare the intranasal liquid film distributions and subregional doses (olfactory, superior meatus, vestibule) between the baseline and optimized delivery systems.

## 2. Materials and Methods

### 2.1. Study Design

An existing nasal airway geometry was tested in this study, which was reconstructed based on MRI images of a 53-year-old male (Figure 1) [34]. A vertex-to-floor (or Mecca) head position was tested so that the liquid film translocation to the olfactory region can be maximized due to the alignment of gravity with the olfactory mucosa (Figure 2a). The influences of different nasal spray parameters were considered (Figure 2b). A flow rate of 1.8 L/min was simulated to present the gentle inspiration that was normally adopted during nasal spray therapies.

The first step was to choose an “optimized” delivery mode that can maximize the initial deposition to or close to the olfactory region. The intranasal distribution of sprays can be affected by factors related to patients (respiration, head position, health condition), devices (droplet size, velocity, plume angle), and administration (or patient–device interaction, i.e., release position and orienting). For a given subject with a given head position, the variables of interest included the spray release position, device orientation, droplet size, velocity, and spray plume angle; among them, different combinations were tested to identify an “optimized” delivery system. Three criteria were employed to judge whether one delivery mode was better than the other, with descending importance: (1) a higher deposition fraction (DF) in the olfactory region; (2) a higher DF in the superior meatus, where deposited sprays can readily translocate into the olfactory region by gravity; (3) a lower deposition in the vestibule, where the deposited medications are wasted. In doing so, the sensitivities of the three criteria to each of the seven parameters were quantified, as listed in Table 1. The device orientation in the transverse plane (i.e., β) was kept at 15°, which represents the device pointing to the outer corner of the eye to maximize the turbinate deposition and minimize the septal deposition. The device orientation in the sagittal plane (i.e., α) was varied in a range of 0–90° to maximize the deposition to the upper nose and minimize deposition in the vestibule. For a given nasal spray device with a given formulation, the plume angle was approximately constant; a range of 10–60° plume angle was considered to test whether the standard nasal device was optimal for olfactory drug delivery, and if not, what was the optimal plume angle, and what percentage could be improved.

In the second step, for the selected delivery system, the liquid film formation and translocation were numerically predicted at different times after application. The spatial and temporal distribution of liquid film, as well as the time variation of the olfactory dose, were quantified.

In the third step, the applied doses (i.e., liquid volume) of the nasal spray were considered in their capacity to enhance the dosing to the olfactory region and their respective delivery efficiency. The nasal spray volume determined whether the liquid film will overflow the initially deposited region and translocate to other regions. With inadequate spray doses, all droplets would be confined where they initially contacted the nasal epithelium because the stabilizing surface tension and viscosity dominated over the destabilizing gravitational component (or self-weight).

### 2.2. Nasal Airway Model

Anatomically accurate nasal airway models are essential for quantitative assessment of the intranasal dosimetry of nasal sprays. An existing nasal airway geometry was used in this study, which had been reconstructed from MRI scans (512 × 512-pixel resolution) of a healthy 53-year-old male [34]. The procedures for model development were briefly described here, with more details presented in Xi et al. [35]. The multi-slice MRI images were segmented using MIMICS (Materialise, Ann Arbor, MI, USA) to convert the raw image data into a set of cross-sectional contours that define the solid geometry. A surface geometry was then built in Gambit (Ansys, Inc., Canonsburg, PA, USA) by patching over these contours. Figure 1a displays the nasal airway model, together with a mouth cavity and face model, and a nasal spray device inserting into the nostril. The development of the mouth and face models were explained in Xi et al. [36]. To quantify the subregional deposition fractions (DFs), especially in the olfactory region and the superior meatus (upper turbinate), the nasal airway surface was divided into different sections that included the nasal vestibule (Vest), nasal valve, turbinate (TR), olfactory region (OL), nasopharynx (NP), and pharynx (Figure 1b). As the nasal spray was often applied to one nose at a time, the liquid film pattern on the septal walls of the treated nasal passage would be blocked by its own turbinate on one side and the untreated nasal passage on the other side. To visualize liquid film distributions (and evolution) on the septal wall, surfaces on the right and left nose were also separated. To reveal the turbinate structure, the right nasal passage was also cut open along the curved ridges of the nasal top and approximately the middle line of the nasal floor (Figure 1b). The pendant-shaped turbinate heads intruded into the nasal cavity, forming the inferior meatus between the nasal floor and the lower turbinate, the middle meatus between the lower and middle turbinate, and the superior meatus between the upper turbinate and nasal roof (Figure 1b, right panel). Computational mesh of the nasal airway was generated using ANSYS ICEM CFD (Ansys, Inc., Canonsburg, PA, USA), with tetrahedral meshes in the volume and fine body-fitted meshes in the near-wall region [37] (Figure 1b, right panel). A mesh sensitivity analysis was conducted and the grid-independent predictions were established at a mesh size of 2.8 million with four layers of near-wall prismatic cells, as shown in Figure 1b. The height of the first-layer cell was 0.05 mm, and the height inflation ratio was 1.3, leading to a total height of 0.31 mm for the four prism layers.

### 2.3. Spray Release Model

Varying delivery modes of the nasal spray were tested with different release positions, application angles (α, β), and spray plume angle (θ) (Figure 2b). The spray application angle relative to the nostril had two degrees of freedom: α in the sagittal plane and β in the transverse plane.

Nasal spray velocities and droplet sizes were modeled following the measurements in the literature, which are approximately 10 m/s and 60 ± 25 µm, respectively, for a typical nasal spray product [18,38,39]. In this study, monodisperse aerosol droplets were generated (Figure 3a). For the aerosols of 60 µm, a count of 884,194 droplets were generated to represent a single dose of 0.10 mL per actuation. The above droplet count was calculated by 0.10 mL divided by the volume of one droplet. For 0.20–0.8 mL applied doses, the droplet count was correspondingly scaled. Based on the consistent observations that all nasal sprays had a jet along the centerline in contrast to slow, diluted droplets in the plume outskirts, a normal distribution in both particle initial positions and existing velocities were implemented, as shown in Figure 3a. Furthermore, a random number within the range of 50% was applied to the local velocities to simulate the variability in the spraying process [18,40].

### 2.4. Airflow and Droplet Transport Models

Incompressible flow and isothermal conditions were assumed in this study. The flow field was simulated using the low Reynolds number (LRN) k–ω model [41]. This model has been validated in many studies to be able to predict accurately the particle transport and deposition in the oral airway [42], nasal cavity [43], and lungs [44]. Moreover, the LRN k–ω model was demonstrated to predict accurately the flow regime transition when the turbulent viscosity approaches zero [45]. Governing equations for the turbulent kinetic energy (k) and dissipation rate (ω) can be found in Wilcox [45]. The behavior and fate of inhaled particles were simulated with a Lagrangian tracking model with user-defined functions [46]. This fluid–particle transport model has been well tested in human respiratory airways [47,48].

### 2.5. Eulerian Wall Film Model

When droplets impinge on the wall, four different mechanisms are possible depending on the droplet impact energy and wall temperature. Figure 3b schematically shows these mechanisms: stick, splash, evaporate, and spread [49]. In this study, droplet evaporation was neglected considering that the body temperature is much lower than the boiling temperature of the sprays (~100 °C).

After droplets are collected on the wall surface, a liquid thin film forms and will move/spread depending on the net forces acting on it. The Eulerian wall film model considers four major processes for the droplet-wall interactions: interaction during the initial impact with a wall boundary, subsequent tracking of the air–liquid interface, calculation of film variables, and coupling to the gas phase and solid wall. The governing equations for the mass and momentum conservation of the wall-film are the following:(1)$$\frac{\partial h}{\partial t}+\nabla_{s}\times \left(h{\vec{V}}_{l}\right)=\frac{{\dot{m}}_{s}}{\rho_{l}}$$
(2)$$\frac{\partial h{\vec{V}}_{l}}{\partial t}+\nabla_{s}\times \left(h{\vec{V}}_{l}{\vec{V}}_{l}\right)=-\frac{h\nabla_{s}\left(P_{g}+P_{h}+P_{\sigma }\right)}{\rho_{l}}+{\vec{g}}_{\tau }h+\frac{3}{2\rho_{l}}{\vec{\tau }}_{fs}-\frac{3\vartheta_{l}}{h}\vec{V}+\frac{\dot{q}}{\rho_{l}}$$

In Equation (1), *h* is the film height, $\nabla_{s}$ is the surface gradient operator, ${\vec{V}}_{l}$ is the mean film velocity, *ρ_l_* is the liquid density, and ${\dot{m}}_{s}$ is the mass source due to droplet collection, film separation, and film stripping. In the right-hand side of Equation (2), the first term represents the loading in the normal direction, the second-to-fourth terms represent the loading in the tangential direction, while the last term is the momentum gain or loss due to droplet collection or separation. Specifically, *Pg* is the air pressure, $P_{h}=-\rho h\left(\vec{n}\times \vec{g}\right)$ is liquid-film-induced pressure normal to the film (spreading), and $P_{\sigma }=-\sigma \nabla_{s}\times \left(\nabla_{s}h\right)\;$is the pressure caused by surface tension. The second term $\left({\vec{g}}_{\tau }h\right)$ is the gravitational effect tangential to the film, the third term $\left(3{\vec{\tau }}_{fs}/2\rho_{l}\right)$ is the viscous shear force at the air–film interface, and the fourth term $\left(3\vartheta_{l}\vec{V}/h\right)$ is the viscous force at the film–wall interface. The last term $\dot{q}$ is the momentum source term, with $\dot{q}={\dot{m}}_{s}\left({\vec{V}}_{p}-{\vec{V}}_{l}\right)$ and ${\vec{V}}_{p},\;{\vec{V}}_{l}\;$being the droplet and liquid film velocities, respectively. To calculate the viscous force, the film velocity is assumed to have a parabolic profile.

### 2.6. Numerical Methods

ANSYS Fluent 19 (Canonsburg, PA, USA) was utilized to solve the governing mass, momentum, energy, and Eulerian liquid film governing equations. There were four steps in total. First, continuous-phase airflows were simulated, with zero ambient pressure at the inlet and constant vacuum pressure at the tracheal outlet to represent a gentle inhalation at 1.8 L/min. The differential pressure between the absolute pressure at the inlet and atmospheric pressure was taken as equal to zero. Second, the temperature field was simulated with inlet temperature at 20 °C and wall temperature at 37.5 °C. In the third step, discrete-phase droplet trajectories were simulated, and their deposition rates and distributions were quantified using in-house user-defined functions (UDFs). Finally, the liquid film formation on the airway surface from deposited droplets and the wall film temporal translocation were simulated and quantified. Airway wall boundary conditions of no-slip condition for airflow, constant temperature for energy, perfect particle adsorption for particle deposition, and zero initial height for wall film were applied in the computational model. The airway wall was also assumed smooth and rigid. Particle size considered ranges from 10 µm to 120 µm, with a volumetric-mean diameter VMD = 60 µm. Convergence sensitivity analyses were performed to establish grid-independent and particle-count-independent results [50]. The final mesh size was 2.8 million, and the final number of test particles was 884,194 to match the dose of 0.10 mL drug content. The convergence criteria of the flow are 1.0 × 10^−5^ for normalized mass and momentum. To track the liquid film motion, variable time steps were used, with the minimum being 1.0 × 10^−7^ s and the maximum being 1.0 × 10^−5^ s to ensure the Courant number less than 1.0. Simulating the liquid film motion for 200 ms required around 200,000 time steps and took around 80 h in an AMD Ryzen 9 3960 24-Core workstation with 256 G RAM and 3.79 GHz frequency).

## 3. Results

### 3.1. Baseline Delivery System

#### 3.1.1. Airflow Field

Figure 4a shows the inspiratory airflow dynamics in the right nasal passage at an inhalation flow rate of 1.8 L/min. Due to the low speed, the flow in the nasal airway is predominantly laminar. The curvature streamlines are smooth in most of the nasal cavity, except in the nasopharynx and oropharynx where a drastic airspace expansion occurs. The airflow has the highest speed at the nasal valve (Figure 4a) and the lowest in the nasopharynx. Two recirculation zones are observed in the nasopharynx close to the ventral and dorsal walls (Figure 4a). The streamlines overall exhibit a U-shape from the nostril to the pharynx. Following one individual streamline, the flow particle that enters the nose at the tip of the nostril goes upward to the superior meatus can potentially reach the olfactory region (i.e., the first streamline in Figure 4a, red arrow). Flow elements that enter the nose slightly below the nostril tip, however, go to the middle meatus, as illustrated by the second streamline in Figure 4a. By contrast, flow elements entering the nose from the base of the nostril go to the inferior meatus. Note that the distance between two adjacent streamlines is directly associated with the flow rate in that region. The larger inter-streamline distance in the superior meatus coincides with its low ventilation rate, while the clustering of streamlines in the middle meatus indicates a much higher ventilation rate, as shown in Figure 4b.

The right panel of Figure 4a shows the velocity contours in a sagittal (y-z) and transverse (x-y) planes. A pear-shaped background (gray) was added behind that represents the tissues and nasal mucosa enclosing the middle meatus. The two round-headed structures are the inferior and middle turbinate, respectively. The mucosa lining the nasal epithelium is rich in capillary blood vessels, whose swelling can significantly reduce the clearance of the already narrow nasal passage, leading to a nasal obstruction that notably changes the airflow and resistance. In the slice considered, the highest velocity (red color) occurs in the middle meatus close to the nasal septum, while low-velocity flows (blue color) occur at the ends of the three meatuses.

#### 3.1.2. Initial Deposition of Nasal Droplets

The initial droplet deposition delivered using the baseline mode is shown in Figure 4c–e. The nasal spray releasing properties (red vectors) are shown in Figure 4c, i.e., droplet size d_p_ = 60 µm, droplet velocity V = 10 m/s, plume angle θ = 45°, device orientation α = 60° from the vertical, and with a 4 mm insertion (green plane) into the right nostril. The nasal spray deposition distribution is shown in Figure 4d, with 50% transparency of the airway wall so that deposited droplets in both walls can be viewed. As expected, the majority of spray droplets are deposited in the front nose, particularly in the nasal valve (Figure 4d). There are also droplets reaching the turbinate region, but only a small portion of them deposited in the superior meatus, and even fewer in the olfactory region (zoomed inset in Figure 4d).

Since a vertex-to-floor head position was taken during spray application, the nasal airway and droplet deposition were replotted upside down to be consistent with the head position (Figure 4e). Droplet accumulations occur on the inferior turbinate (red dotted circle) and the lower nasal valve (green arrow). Deposition in each subregion of the nose were quantified: 32.1% in the vestibule, 55.9% in the nasal (upper panel); 5.7% in the inferior meatus, 5.2% in the middle meatus, 1.0% in the superior meatus, and 0.1% in the olfactory region (Figure 4e). This extremely low olfactory DF represents a challenge for effective nose-to-brain drug delivery. Note that the upside-down head position aligns the olfactory region with the gravity, which is hypothesized to have given optimal dosing to the olfactory region.

### 3.2. Input Sensitivity Studies

#### 3.2.1. Spray Release Position Effect

To improve the olfactory dispensing, different delivery parameters were explored, including the spray releasing position, spray orientation, droplet size, droplet velocity, and plume angle. Figure 5a shows the spray releasing position effects on the deposition distribution. Seven positions were considered that included three nozzle insertion depths and three points at certain depths (L2 and L3). Only one point (middle) was considered at the 8 mm insertion (L1) because a 60°-oriented nozzle intruding to the tip or base of the L1 plane interfered with the vestibule walls, rendering it infeasible to do so without hurting the nose. Marked differences in the spray deposition patterns are noted among the seven release positions; however, the OL dispensing is low for all. The slit-like nasal valve, as well as the narrow, convoluted nasal TR nasal passage, defies the intuitive hypothesis that pointing the nasal spray to OL or aligning gravity with OL will optimize the OL dosing. The interplay between the labyrinthic nasal cavity and the high-speed spray plume cannot be straightforwardly envisioned and has to be examined using numerical modeling and simulations. For high-speed nasal sprays with a mean droplet size of 60 µm, inertial impaction and interception are the two dominant deposition mechanisms. Gravitation can also be important in regulating the trajectories of the spray droplets.

Comparison of subregional DFs in the olfactory (OL) region, upper turbinate (TR), and vestibule (Vest) is shown in Figure 5b. The criteria for an improved delivery scenario include (1) higher DFs in the OL and upper TR and/or (2) reduced DF in the nasal vestibule. As a result, the spray release position at L2b (4 mm insertion at the middle of the plane) was selected based on its second-highest DF in OL and upper TR and lowest DF in the Vest. By comparison, L3b was not selected because of its high vestibule waste despite a high OL dose.

#### 3.2.2. Nasal Spray Application Angle (α) Effect

The effect of the spray application angle, α ranging from 0° to 90° is shown in Figure 6, with baseline spray properties (d_p_ = 60 µm, V = 10 m/s, and θ = 45°) and a spray release position at L2b. The high sensitivity of the spray deposition pattern to the application angle (or device orientation) is obvious. While a vertical application (α = 0°) leads to all droplets being deposited in the nasal vestibule and valve, a horizontal application (α = 90°) leads to a dominant deposition in the floor of the nasal valve and turbinate. Both application angles of 30° and 60° manage to dispense aerosol droplets to the OL proximity. In addition, the fan-shaped plume of both angles covers the OL.

Comparison of the subregional DF among seven application angles (α = 0°, 15°, 30°, 45°, 60°, 75°, 90°) in the OL, upper TR, and Vest is shown in Figure 6b. It is observed that α = 45° gives rise to the highest doses in both the OL and upper TR, while the lowest vestibular waste occurs at α = 60° and the second lowest at α = 45°. As a result, the application angle of 45°, together with the L2b release position, is used for olfactory delivery.

#### 3.2.3. Effects of Nasal Spray Properties

To optimize the nasal spray system for OL drug dispensing, the influences from the nasal spray properties (i.e., droplet size, velocity, and plume angle) were also examined. Figure 7a,b shows the spray deposition distribution for different droplet sizes (d_p_: 10–120 µm) and spray plume angles (θ: 10–60°), respectively. In Figure 7a, droplets of 30 µm and smaller can still be affected by the airflow field, leading to different deposition patterns (the first two panels of Figure 7a). Droplets larger than 30 µm, however, are predominantly controlled by inertia and gravity, and their deposition distributions are largely the same (the last two panels of Figure 7a). Large differences result from the spray plume angle θ, with a very narrow plume delivering more focused doses, in contrast to a wide plume delivering diffusive doses and covering a more extensive region (Figure 7b).

Considering that at α = 60°, the narrow spray missed most of the olfactory region (Figure 7b), different application angles (α = 30°–45°) were tested again, but this time for a spray with θ = 10°, as displayed in Figure 7c. Results show that a certain amount of droplets reach the olfactory region at α = 45°, while negligible droplets do so at α = 30°, 35°, and 40°with the 30° spray, missing the OL and the 35–40° sprays being filtered out before reaching the OL (Figure 7c).

#### 3.2.4. Identifying the Optimal Delivery System

A summary of the subregional DFs in all of the above test cases is presented in Figure 8, including the parametric study of the droplet size (third group), droplet velocity (fourth group), spray plume angle (fifth group), as well as application angle (sixth group). It is shown that the subregional DF has the highest sensitivity to the plume angle (fifth group of Figure 8), where much-elevated deposition in the OL and upper TR are observed for θ = 10° and 20° (i.e., narrow sprays), while the 20° plume generates a higher OL DF than the 10° plume (0.55% vs. 0.37%), and the 10° plume gives rise to a much higher DF in the upper TR (5.54% vs. 2.97% at θ = 20°). Considering that the deposited droplets can form liquid film and translocate under gravity, the 10° plume is chosen over 20° plume for later OL deliveries by hypothesizing that the droplets in the upper TR can move into the olfactory region.

Further testing to identify the optimal application angle for the 10° plume spray reveals that α = 45° gives the highest DF in both the OL and upper TR, as well as negligible waste in the vestibule (sixth group in Figure 8). As a result, the proposed nasal spray delivery system has a release position at L2b, an application angle of 45°, and a plume angle of 10°, while the subject takes a vertex-to-floor head position. The droplet size and velocity, when larger/higher than 30 µm and 7.5 m/s, have a much lower impact on the OL and upper TR deposition (third and fourth groups in Figure 8). Note that the “optimal” delivery system is the best among all test cases considered in this study, as listed in Table 1. This was not obtained using an optimization algorithm with objective function and constraints, which would require a much large number of test cases and thus prohibitively large computational resources in this study.

### 3.3. Wall Film Translocation

#### 3.3.1. Wall Film Intranasal Distribution at a 0.1 mL Dose

The optimized nasal spray delivery system was subsequently used to test the hypothesis that gravitational translocation may significantly enhance the nasal spray dosimetry in the olfactory region. The process of liquid formation from deposited droplets as a function of time is shown in Figure 9a after applying a spray dose of 0.1 mL. For comparison purposes, the initial droplet deposition is displayed in Figure 9b.

The liquid film first forms on both sidewalls of the nasal valve, where droplets quickly deposit in a large fraction (t = 0.02 ms, Figure 9a). It is noted that not all deposited droplets are perceivable in this figure considering the color depth is proportional to the droplet accumulation. With more droplets deposited in the nose, the liquid film grows in both the extent and intensity (depth), as shown in Figure 9a, at t = 2.0 ms and 200 ms. The film pattern shows negligible variation at t = 1.0 s and 2.0 s (not shown for lack of variation). This is because that the limited spray dose (0.1 mL in this case) leads to a finite thickness of the liquid film, whose self-weight is inadequate to overcome the stabilizing surface tension. In other words, the applied dose is not enough to overflow the nasal epithelium where the droplets are deposited. It is also noted that the final liquid pattern is similar to but not the same as the droplet deposition (Figure 9b vs. Figure 9a). The deviation is presumably because of the liquid spreading from high local droplet accumulations. On the other hand, there is no apparent liquid film in the upper TR and OL because of the much lower doses (fewer particles per unit area) than in the nasal valve.

#### 3.3.2. Wall Film Intranasal Distributions at Increased Doses (0.2–0.8 mL)

Nasal spray delivery with the optimized system and increasing applied doses is shown in Figure 10a. At 0.2 mL; there is no apparent droplet translocation into the olfactory region, indicating that the applied dose is still under the capacity of the nasal epithelium to hold liquid. Doubling the applied dose (0.4 mL) increases both the extent and intensity (thickness) of the liquid film. The migration of the liquid film also starts on the septal wall, reflective of the start of liquid overflow as the thickness of the liquid film builds up. Further increasing the applied dose to 0.6 mL substantially enhances the overflow and liquid film migration into the OL following the gravitational direction. This is because all added 0.2 mL will overflow and migrates to surrounding regions, either moving downward driven by gravity or spreading laterally driven by the capillary force (surface tension) and confined (or guided) by the nasal geometry.

Interestingly, the increase of the OL dose with applied spray dose is not linear, as shown in Figure 10b, with a slow increase before 0.4 mL, a quick increase between 0.4 and 0.7 mL, and a slow increase again for doses larger than 0.7 mL. This nonlinearity is speculated to result from a rapid increase of the film migration in the lateral direction than the downward direction. Note that increasing the applied spray dose beyond 0.6 mL can still increase the absolute OL dosage (Figure 10b); however, the benefit dwindles, as the deposition fraction starts to drop for applied doses larger than 0.6 mL (Figure 10c). This is because the more liquid film moves to the back of the nose rather than to the OL region when the applied dose is more than 0.6 mL, as illustrated in the third and fourth panels of Figure 10a.

#### 3.3.3. Wall Film Intranasal Distributions with the Baseline Delivery System

For comparison purposes, nasal sprays administered with the baseline system (i.e., release position at L2b, α = 60°, d_p_ = 60 µm, V = 10 m/s, θ = 45°, as listed in Table 1) were also considered with applied doses ranging from 0.2 mL to 0.8 mL (Figure 11). The initial droplet deposition distribution is shown in Figure 4. Several differences in the liquid film patterns are noteworthy (Figure 11 vs. Figure 10). First, fewer signs of liquid film dripping are observed in the deposition patterns using the baseline delivery system than those with the optimized system. Note that the optimized system uses a narrow spray plume (θ = 10°), while the baseline uses a wider one (θ = 45°). The more localized droplet deposition from a narrow spray plume is more prone to overflow the local epithelial liquid-holding capacity and elicit liquid dripping. By comparison, droplet deposition from a wide-angle plume is more scattered throughout the nasal cavity and thus is more difficult to overflow the epithelial holding capacity. Thus, the migration of liquid film does not rely on dripping but on capillary-force-driven spreading.

Second, the enhancement to the OL dosage due to film migration (or droplet translocation) appears much lower using the baseline system than that using the optimized system. At an applied dose of 0.6 mL, the baseline OL DF is 1.31%, while the optimized OL DF is 6.23% (Table 2).

Third, the drug waste in the front nose differs significantly. The drug waste in the vestibule (Vest) is much higher using the baseline system than the optimized system and is getting worse with increasing applied doses (green-square line). The drug waste in the valve is also higher using the baseline system but does not change much with increasing applied doses from 0.2 mL to 0.8 mL (blue-diamond line, Figure 11b), reflecting the wall film saturation in the nasal valve at 0.2 mL applied dose, with extra deposited droplets being translocated to the turbinate region. Indeed, the TR dosage increases more than linearly with the applied dose (red-triangle line, Figure 11b and a positive DF slope in Figure 11c), with an influx of liquid film from the nasal valve.

## 4. Discussion

In this study, a computational testing platform for nasal spray applications was developed that considered the patient-, device-, and administration-related variables, including the patient-specific geometry, respiration rate, head position, the nasal spray droplet size, velocity, and plume angle, as well as the nozzle insertion and orientation relative to the nostril in both planes. This platform has been used to identify the optimized delivery system that maximizes the initial dose in and close to the olfactory region and minimizes the drug loss in the vestibule, which was further used to simulate the wall film formation and translocation after spray applications.

The results of this study corroborate the previous finding that nasal spray droplets predominantly deposit in the anterior nose, i.e., the vestibule and nasal valve. Using a radial-hole inhaler, Herranz González-Botas et al. [10] observed no dye deposition at the olfactory cleft and middle turbinate but abundantly in the nasal vestibule, nasal septum, and inferior turbinate. Similar observations have been reported by Guo et al. [32], who tested different visualization methods (isotopes, dyes, endoscopy) in nasal hollow casts, and Xi et al. [51] who tested four types of nasal sprays in the sectional nasal casts, where only a small fraction of applied nasal spray penetrated beyond the nasal valve. Such results are expected considering that the spray droplets will deposit on any surfaces that are on their way due to their high inertia and straight trajectories. The vestibule is an approximate 90° bend with converging cross sections, and the nasal valve is a slender flow-limiting slit, both of which are difficult for the high-speed large droplets with radiating trajectories to maneuver through. Likewise, to decrease the drug loss in the front nose, the nasal spray should be released from an appropriate position to avoid vestibule filtration and should have a narrow plume to reduce the nasal valve impaction.

The spray plume angle was found to have a significant influence on the initial olfactory (OL) dose, as well as on the intranasal deposition distribution. For the airway considered in this study, the optimal spray plume angle was identified to be 10°, which is much narrower than most conventional nasal spray devices. This is consistent with Cheng et al. [12], who reported that narrow spray plume angles allowed a higher fraction of droplets to pass the nasal valve. A narrow spray plume also gave rise to more concentrated deposition and thus a thicker liquid film, which could more easily go beyond the epithelium liquid-holding threshold and start to move to other regions. By contrast, a wider plume will result in dispersed deposition and thinner liquid films, which need more spray doses to reach the threshold. Based on the above two reasons (i.e., higher OL dose and easier to move), a narrow plume (i.e., 10°) is recommended over the conventional nasal devices, whose plume angles typically range from 20° to 50° [12,38,52].

The results of this study demonstrated that the liquid film migration after nasal spray application significantly enhanced the OL dosage, as listed in Table 2, which is a summary of Figure 8, Figure 10, and Figure 11. The increase was 8.5 folds (i.e., from 0.16% to 1.31%) for the baseline and 9.4 folds (i.e., from 0.67% to 6.23%) for the optimized case. The drug waste in the baseline case also decreased from 9.40% to 3.74%, indicating the liquid film overflow and translocation into other regions. On the other hand, the vestibule loss in the optimized case increased from 0 to 0.6%, reflecting the film spreading from the adjacent nasal valve. The benefit of optimizing the initial intranasal spray distribution is also shown in Table 2. Relative to the baseline, the OL dose in the optimized case increased by 377% (i.e., from 1.31% to 6.23%), while the drug loss in the vestibule decreased by 84% (i.e., from 3.74% to 0.60%, Table 2).

Limitations of this study include a single airway geometry, one head position, limited test cases of nasal applications, no inter-droplet interactions, monodisperse droplet size distribution, and neglecting the effects of evaporation/condensation, electrostatic charges, and rheological properties. As the intranasal distribution of spray droplets is sensitive to the nasal anatomical details, the olfactory delivery efficiency can be different from the results presented here (i.e., 6.23% in the right passage). The intra-subject and inter-subject variability should be quantified in future studies. However, based on the observation of a ninefold increase or so for both the baseline and optimized cases (Table 2), it is speculated that a similar OL dose enhancement can be expected for other cases. Second, different head positions have been proposed in olfactory dosing besides the vertex-to-floor or Mecca position, such as the upright-head-back position, lying-head-back (or Mygind) position, and Kattecki position (lying on one side with neck turned and chin lifted) [28,29,30,31,53,54], whose effect on OL dose enhancement is unclear. Thirdly, the “optimized” system of this study was based on a limited number of test cases, not an exhaustive peak search in the design space. The significance of initial intranasal dose distribution on final doses, however, has clearly demonstrated using the baseline and optimized systems. The evaporation/hygroscopic effect on droplet dynamics was neglected considering that the temperature is constant and the relative humidity is close to saturation [55]. Additionally, droplet size distribution [56], electrostatic charge [57,58,59,60], and droplet collisions [61] were neglected for simplicity purposes in modeling and simulations. Practically, neglecting these minor factors allowed the isolation of the dominant factors (wall film migration) to be examined in a controlled manner. In addition, these factors only affect the initial droplet deposition, not the subsequent translocation.

## 5. Conclusions

In summary, a computational model for nasal spray applications was developed. Nasal spray droplet deposition and subsequent wall film migration were numerically simulated with a baseline (representing conventional nasal spray devices) and an optimized delivery system in a nasal airway model taking a vertex-to-floor position. Specific findings are as follows.
(1)Nasal spray droplets from the baseline delivery system predominately deposit in the vestibule and nasal valve;(2)Intranasal distribution of initial droplet deposition is highly sensitive to the spray plume angle and device orientation relative to the nostril;(3)The optimal delivery parameters for olfactory dispensing in this study include a spray plume angle of 10° and a device orientation of 45°;(4)Liquid wall film migration with a vertex-to-floor head position enhances the olfactory dosages by ninefold or so, in comparison to the initial olfactory dose for both systems;(5)A delivery rate of 6.2% to the olfactory region is achieved using the proposed nasal spray delivery system and taking a vertex-to-floor head position.

## Figures and Tables

**Figure 1 pharmaceutics-13-00903-f001:**
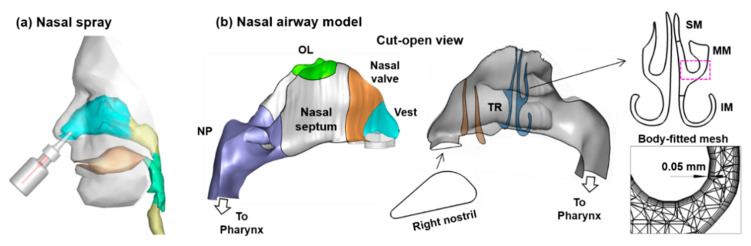
Nasal spray modeling: (**a**) diagram of nasal spray application and (**b**) MRI-based nasal airway model with a cut-open view to reveal the nasal septum and turbinate (conchae) in the right passage. To characterize intranasal spray distribution, the nasal cavity is divided into different sub-regions: vestibule (Vest), nasal valve, turbinate (TR), olfactory region (OL), and nasopharynx (NP). The airspace in the TR region is termed meatus, which is divided into superior meatus (SM), middle meatus (MM), and inferior meatus (IM). Fine prismatic mesh is generated in the near-wall region with the first layer height being 0.05 mm.

**Figure 2 pharmaceutics-13-00903-f002:**
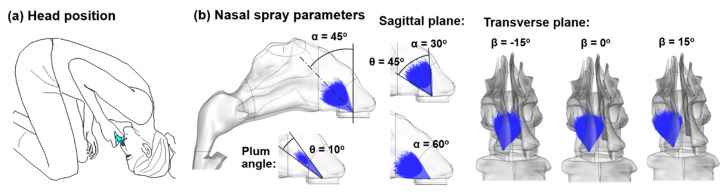
Nasal spray administration: (**a**) vertex-to-floor position and (**b**) different application angles (α, β) and spray plume angles (θ). The spray application angle relative to the nostril has two degrees of freedom: α in the sagittal plane and β in the transverse plane.

**Figure 3 pharmaceutics-13-00903-f003:**
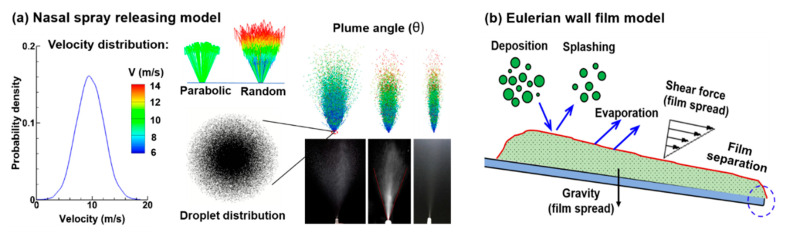
Computational models: (**a**) spray releasing model with prescribed droplet velocity and droplet distributions (normal distributions herein), and varying spray plume angles and (**b**) Eulerian wall film model considering droplet deposition, splashing, stripping, film separation, and film migration due to shear and gravity.

**Figure 4 pharmaceutics-13-00903-f004:**
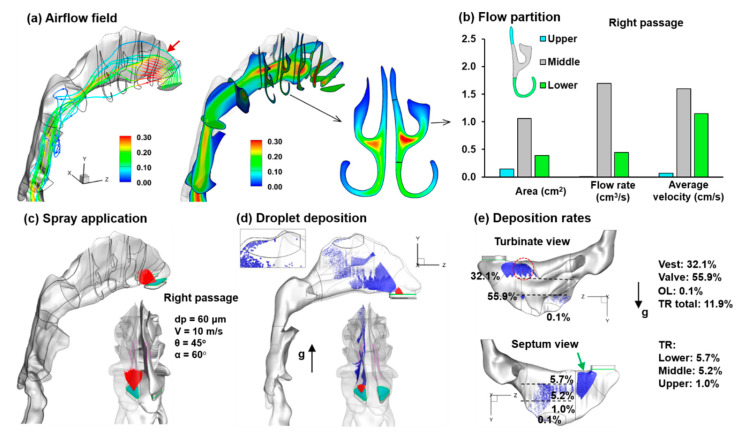
Velocity and spray dosimetry at the baseline condition (i.e., the device orientation α = 60° from the vertical): (**a**) streamlines and the midplane velocity contour at 1.8 L/min; (**b**) flow partition among the superior, middle, and inferior meatus; (**c**) nasal spray releasing, with the droplet size d_p_ = 60 µm, droplet velocity V = 10 m/s, plume angle θ = 45°, and the releasing position being 4 mm into the nostril; (**d**) the initial particle deposition, with very few particles deposited in the olfactory region; (**e**) subregional deposition fractions: 32.1% in the vestibule and 55.9% in the nasal (upper panel); 5.7% in the inferior meatus; 5.2% in the middle meatus; 1.0% in the superior meatus; 0.1% in the olfactory region.

**Figure 5 pharmaceutics-13-00903-f005:**
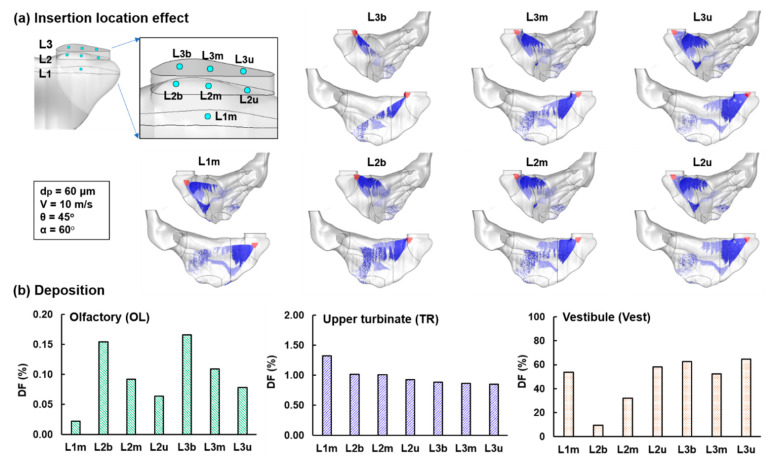
Spray release position effect: (**a**) spray deposition distributions (50% wall transparency) with seven release positions with three insertion depths and three points (tip, middle, base) on certain depths and (**b**) comparison of subregional DF in the olfactory (OL) region, upper turbinate (TR), and vestibule (Vest).

**Figure 6 pharmaceutics-13-00903-f006:**
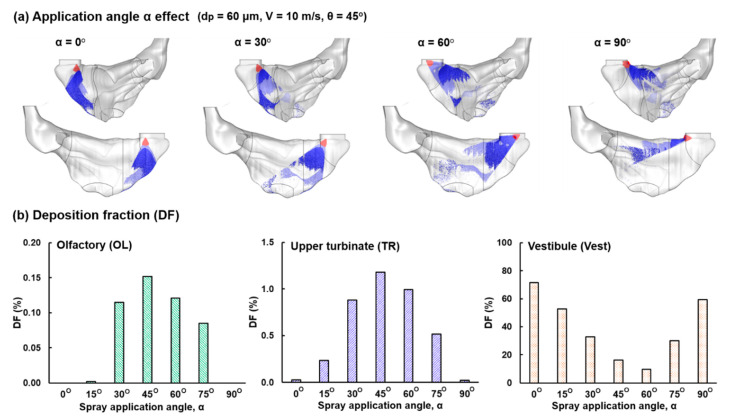
Application angle (α) effect with a spray release position at L2b (a 4 mm insertion from nostril base): (**a**) deposition distributions for α = 0° (vertical), 30°, 60°, and 90° and (**b**) subregional DFs in the OL, upper TR, and Vest vs. α.

**Figure 7 pharmaceutics-13-00903-f007:**
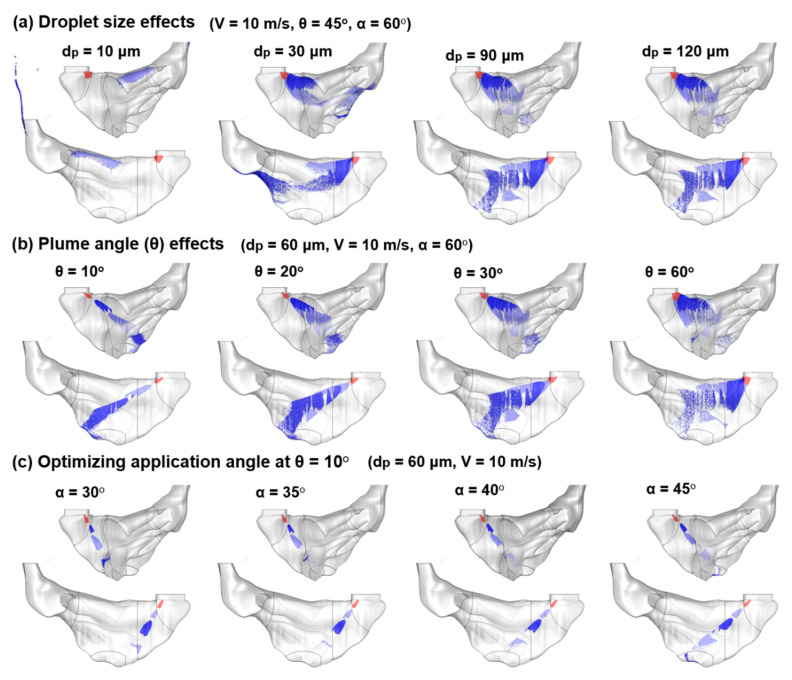
Deposition distribution of spray droplets with different properties: (**a**) droplet size, d_p_: 10–120 µm; (**b**) spray plume angle, θ: 10°–60°; (**c**) application angle, *α* in the range of 30°–45° at the selected plume angle, θ = 10°.

**Figure 8 pharmaceutics-13-00903-f008:**
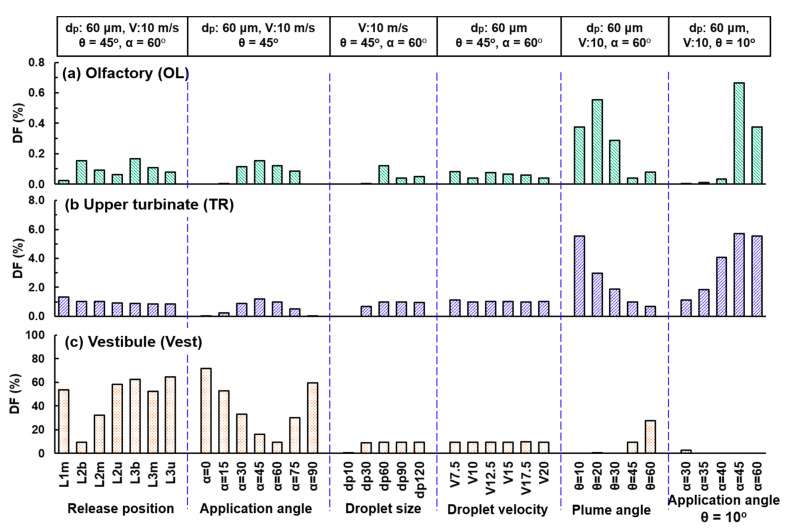
Comparison of the subregional DF among varying delivery scenarios in the (**a**) OL, (**b**) upper TR, and (**c**) Vest.

**Figure 9 pharmaceutics-13-00903-f009:**
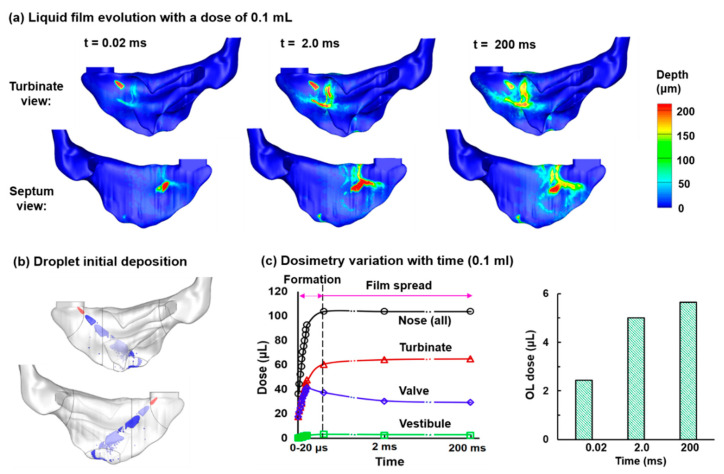
Nasal spray droplet deposition and wall film formation with the proposed delivery system: (**a**) liquid film temporal evolution with a dose of 0.1 mL; (**b**) initial droplet deposition distribution; (**c**) dose variation with time in the nose and OL.

**Figure 10 pharmaceutics-13-00903-f010:**
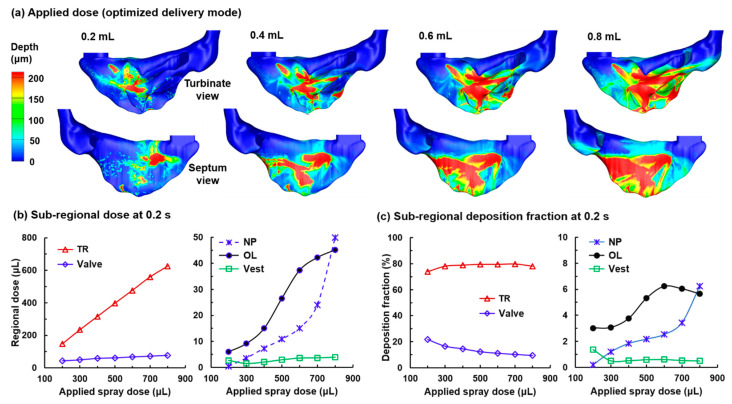
Nasal spray delivery with different applied doses and the optimal delivery system: (**a**) liquid film distribution at 0.2 s after applying a spray dose of 0.2 mL, 0.4 mL, 0.6 mL, and 0.8 mL; (**b**) subregional dosages in the TR, valve, NP, OL, and Vest as a function of applied spray dose; (**c**) subregional deposition fraction in the TR, valve, NP, OL, and Vest as a function of applied spray dose.

**Figure 11 pharmaceutics-13-00903-f011:**
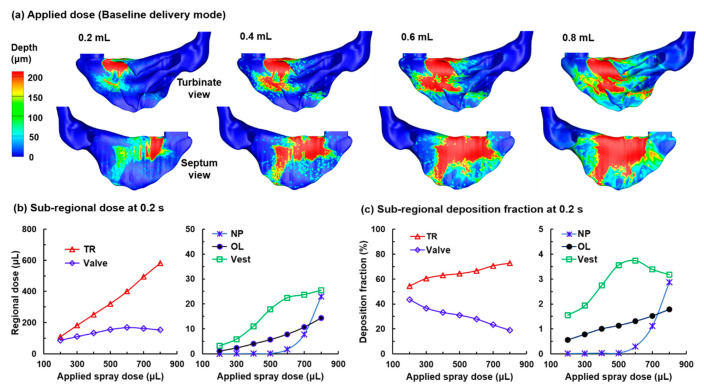
Nasal spray administration with the baseline delivery mode (i.e., α = 60°, θ = 45°, as shown in Figure 4 and Table 1): (**a**) liquid film distribution at 0.2 s after applying a spray dose of 0.2 mL, 0.4 mL, 0.6 mL, and 0.8 mL; (**b**) subregional dosages in the TR, valve, NP, OL, and Vest as a function of applied spray dose; (**c**) subregional deposition fraction in the TR, valve, NP, OL, and Vest as a function of applied spray dose.

**Table 1 pharmaceutics-13-00903-t001:** Nasal spray delivery factors.

	Delivery Mode (Device–Patient Interaction)				Nasal Spray Properties		
Factor	Insertion (mm)	Nostril Positions	Sagittal angle, α (°)	Axial angle, β (°)	Size d_p_ (µm)	Velocity V (m/s)	Plume angle γ (°)
Range	0, 4, 8	3	0–90	15	Mean: 60 15–150	Mean: 10 2–20	10–60
Baseline	3	Middle	60	15	60	10	45
Optimized	3	Middle	45	15	60	10	10

**Table 2 pharmaceutics-13-00903-t002:** Comparison of intranasal dose distributions between baseline and optimized delivery system before and after wall film migration.

Delivery System	Initial Dose			Dose with Film Migration	
	OL	Upper TR	Vest	OL	Vest
Baseline (%)	0.15	1.02	9.40	1.31	3.74
Optimal (%)	0.67	5.70	0.0	6.23	0.60

## Data Availability

The data presented in this study are available on request from the corresponding author. The data are not publicly available due to privacy.

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
