# Peer review of "Liquid Film Translocation Significantly Enhances Nasal Spray Delivery to Olfactory Region: A Numerical Simulation Study"

_pharmaceutics, 2021, doi:10.3390/pharmaceutics13060903_

Round 1

Reviewer 1 Report

see attached file

Author Response

We have carefully considered each of the Reviewer's comments and revised the manuscript accordingly.  The point-by-point responses can be viewed in the attached PDF file.   

Reviewer 2 Report

This paper by Si. X.A. et al. is a very good example of fundamental studies in the pharmaceutical field to improve nose-to-brain delivery strategies. The experimental design is rationally chosen, and the derived results should prove important for the scientific community and hopefully pave the road towards potentially innovative and effective approaches in nasal delivery strategies.

I recommend for this manuscript to be published as is.

Author Response

This paper by Si. X.A. et al. is a very good example of fundamental studies in the pharmaceutical field to improve nose-to-brain delivery strategies. The experimental design is rationally chosen, and the derived results should prove important for the scientific community and hopefully pave the road towards potentially innovative and effective approaches in nasal delivery strategies. I recommend for this manuscript to be published as is.

Response: We highly appreciate the Reviewer’s strong support for this study.

Reviewer 3 Report

The paper entitled “Liquid Film Translocation Significantly Enhances Nasal Spray  Delivery to the Olfactory Region: A Numerical Study” by Si et al. describes a computational testing platform for nasal spray application. The authors state clearly for the objectives of their study and describe a well-organized study design. Moreover, in the discussion section the authors highlight the limitations of their study and propose additional variables which should be taken into account in future studies.

I suggest only some minor revisions.

Use standard symbol of angle degree.

What does it mean “90o-bend” line 534

Avoid slang language like “cached” line 48

In the description of “Eulerian Wall Film Model” When droplets impinge the mechanisms following droplet impinging in the text are four:  stick, rebound, spread, and splash, while in caption 3b they seem to be different ( plashing, stripping, film separation, and  film migration due to shear and gravity). It is better that the authors explain the mechanisms considered in a better and unambiguous way.

I will change the title in Liquid Film Translocation Significantly Enhances Nasal Spray 2 Delivery to the Olfactory Region: A Numerical Simulation Study

Author Response

Use standard symbol of angle degree.
Response: All angle degree symbols have been corrected (e.g.: 90o to 90o). These format errors happened when I copied the text into the journal template. 

What does it mean “90o-bend” line 534
Response: It should be “90o-bend”, which is an L-shaped bend.

Avoid slang language like “cached” line 48
Response: “cached” was changed to “located”.

In the description of “Eulerian Wall Film Model” When droplets impinge the mechanisms following droplet impinging in the text are four:  stick, rebound, spread, and splash, while in caption 3b they seem to be different ( plashing, stripping, film separation, and film migration due to shear and gravity). It is better that the authors explain the mechanisms considered in a better and unambiguous way.
Response:  We appreciate the Reviewer’s careful review. The sentence was changed to be consistent with Fig. 3b: “Figure 3b schematically shows these mechanisms: stick, splash, evaporate and spread.” 

I will change the title in Liquid Film Translocation Significantly Enhances Nasal Spray 2 Delivery to the Olfactory Region: A Numerical Simulation Study
Response: the title was changed as suggested.